# Cortico-hippocampal network connections support the multidimensional quality of episodic memory

Rose A Cooper*, Maureen Ritchey

Department of Psychology, Boston College, Boston, United States

**Abstract** Episodic memories reflect a bound representation of multimodal features that can be reinstated with varying precision. Yet little is known about how brain networks involved in memory, including the hippocampus and posterior-medial (PM) and anterior-temporal (AT) systems, interact to support the quality and content of recollection. Participants learned color, spatial, and emotion associations of objects, later reconstructing the visual features using a continuous color spectrum and 360-degree panorama scenes. Behaviorally, dependencies in memory were observed for the gist but not precision of event associations. Supporting this integration, hippocampus, AT, and PM regions showed increased connectivity and reduced modularity during retrieval compared to encoding. These inter-network connections tracked a multidimensional, objective measure of memory quality. Moreover, distinct patterns of connectivity tracked item color and spatial memory precision. These findings demonstrate how hippocampal-cortical connections reconfigure during episodic retrieval, and how such dynamic interactions might flexibly support the multidimensional quality of remembered events.
DOI: https://doi.org/10.7554/eLife.45591.001

## Introduction

Memories for past events are highly complex, allowing us to travel back in time and subjectively re-experience episodes in our lives. These events are not stored and played back to us as we experienced them; rather, they are reconstructed in a hierarchical manner. Episodic reconstruction is thought to be facilitated by hippocampal-neocortical processes that rebuild the rich content and quality of past events within a spatio-temporal framework (*Barry and Maguire, 2019*; *Ranganath, 2010*; *Ritchey et al., 2015a*; *Robin, 2018*) and integrate them with prior knowledge (*Morton et al., 2017*). In turn, this adaptive, reconstructive process can lead to forgetting of specific event features and variability in the precision with which different features are remembered (*Schacter et al., 2011*).

Previous research has found widespread increases in cortical and subcortical brain activity when people successfully remember rather than forget events (*Rugg and Vilberg, 2013*). Beyond changes in activity, large-scale brain networks increase their communication strength during episodic retrieval tasks (*Fornito et al., 2012*; *Robin et al., 2015*; *Westphal et al., 2017*), where functional connectivity, particularly of the hippocampus, is increased when events are remembered compared to forgotten (*Geib et al., 2017a*; *King et al., 2015*; *Schedlbauer et al., 2014*; *St Jacques et al., 2011*). Such neural changes are validated by behavioral evidence showing that event features are dependent on one another in memory, emphasizing that remembering involves the binding of distinct elements into a single, coherent event representation (*Horner and Burgess, 2013*; *Horner and Burgess, 2014*). This binding process is widely thought to be facilitated by the hippocampus (*Barry and Maguire, 2019*; *Horner et al., 2015*; *Moscovitch et al., 2016*; *Ritchey et al., 2015a*). Therefore, episodic retrieval is likely dependent on the coordination of memory 'hubs' such as the hippocampus

*For correspondence:
rose.cooper@bc.edu

Competing interests: The authors declare that no competing interests exist.

with neocortical regions to reconstruct and integrate the diverse components of memory representations.

Despite this research, little is known about how changes in hippocampal-cortical communication flexibly support the multidimensional quality of remembered events. Distinct cortical areas support the different building blocks of episodic memory: for instance, parahippocampal cortex (PHC) is thought to provide the hippocampus with spatial context information, whereas perirhinal cortex (PRC) codes for items within this context (*Davachi, 2006*; *Diana et al., 2007*; *Diana et al., 2010*; *Staresina et al., 2013*; *Staresina et al., 2011*). Moreover, these medial temporal cortical regions are situated within two large-scale networks (*Ranganath and Ritchey, 2012*; *Ritchey et al., 2015a*). These networks show functional separation but some common hippocampal connections (*Kim et al., 2018*; *Libby et al., 2012*; *Ritchey et al., 2014*; *Wang et al., 2016*), and have been proposed to support complementary memory functions. The PHC is part of a posterior-medial (PM) system thought to form situation models of events (*Ritchey et al., 2015a*). PM regions include retrosplenial cortex, which also demonstrates representational specificity for spatial environment (*Epstein, 2008*), posterior cingulate, precuneus, and angular gyrus, which are recruited during subjectively vivid recollection and represent precise episodic context information (*Baldassano et al., 2017*; *Kuhl and Chun, 2014*; *Richter et al., 2016*; *Robin and Moscovitch, 2017*; *Sreekumar et al., 2018*). In turn, the PRC is part of an anterior-temporal (AT) system supporting item and emotional associations (*Ritchey et al., 2015a*). Within this system, the amygdala binds item-specific features with emotion (*Kensinger et al., 2011*; *Yonelinas and Ritchey, 2015*), and anterior ventral temporal cortex and lateral orbitofrontal cortex are further involved in processing object representations and the affective significance of items to inform decision making and memory (*Libby et al., 2014*; *Rolls and Grabenhorst, 2008*).

A core tenet of the PMAT framework is that cortical systems must interact with each other and with the hippocampus to support the multidimensional nature of episodic memory. However, several aspects of this crucial principle have yet to be directly tested: First, how do functional network connections reorganize during episodic retrieval? Second, do changes in these connections relate to the amount and quality of information bound within memory? Finally, do different patterns of network connectivity changes support the fidelity of different types of memory content? In this study, we tested these questions to determine how cortico-hippocampal networks flexibly coordinate the reconstruction of complex events.

The contribution of network interactions to the phenomenology of memory has been difficult to establish in part due to the nature of memory tests commonly used in conjunction with functional connectivity methods, which have typically relied on binary measures of 'successful' retrieval or subjective ratings of vividness. These methods are insensitive to the diversity of integrated content and objective precision of retrieved events. To this end, we tested participants on a memory reconstruction task to obtain continuous measures of different episodic memory features (*Brady et al., 2013*; *Harlow and Yonelinas, 2016*; *Nilakantan et al., 2017*; *Richter et al., 2016*). Participants learned a series of objects, each with a color, scene location, and emotion association, and then reconstructed the visual appearance of the objects later on. Here, they selected a color from a continuous spectrum and moved around 360° panorama scenes to place the object in its original location, providing a sensitive, objective, and naturalistic way of assessing memory (cf. *Serino and Repetto, 2018*). We predicted that PM and AT systems would show a distinct network structure during encoding, but that, crucially, these networks would become more integrated during episodic retrieval. Moreover, we expected that increased inter-network and hippocampal connectivity would dynamically track binding and the composite quality of features within memory. In line with the representational organization of the PMAT framework, we finally predicted that functional connectivity of PM and AT systems would track memory precision for spatial context and item information, respectively.

## Results

Participants completed an episodic memory task in which they learned three features associated with trial-unique objects: a color from a continuous spectrum, a location within a panorama scene, and an emotionally negative or neutral sound (*Figure 1A*). In a subsequent test, participants were first cued to covertly retrieve as much information about each object as possible, and then they dynamically reconstructed each object's color and scene location (*Figure 1B*), providing continuous

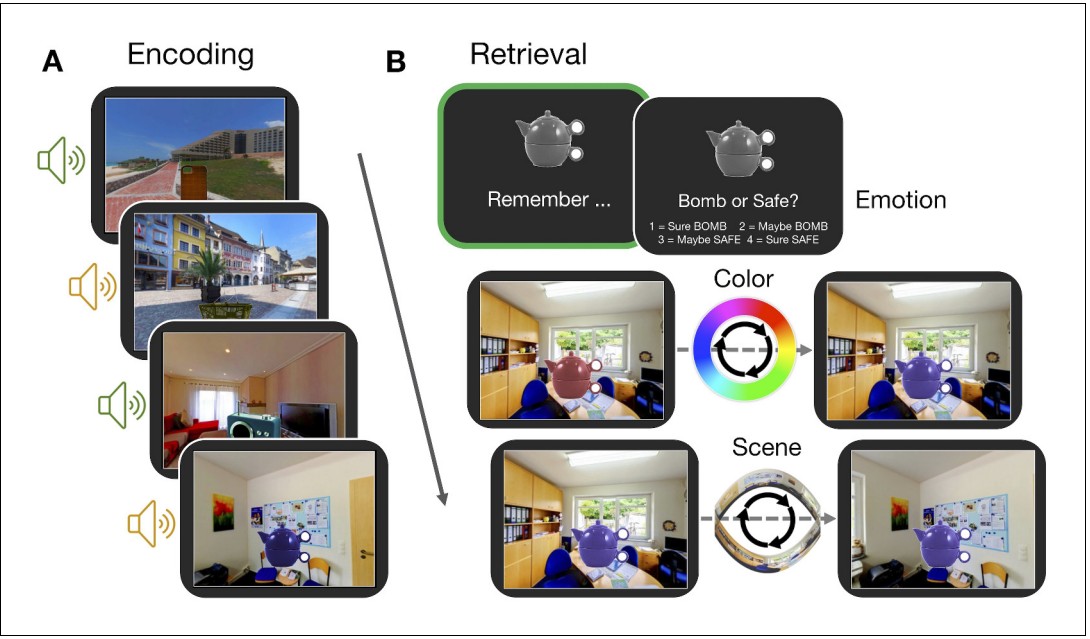

**Figure 1.** Experiment paradigm. (**A**) Participants encoded a series of objects, presented in a specific color and scene location and accompanied by either an emotionally negative (orange; 'bomb') or neutral (green; 'safe') sound. (**B**) For each trial in the memory test, participants first retrieved all features associated with an object in their mind ('remember' event; green box). This remember event was the basis of all retrieval-related fMRI analyses. Participants then retrieved the individual features of the object sequentially. For questions about the color and scene location, participants recreated the object's appearance by moving around the 360° color spectrum and panorama scene. Accuracy was measured in terms of error (response - target). Background panoramic images taken from the SUN 360 database (*Xiao et al., 2012*); objects taken from the Vision and Memory Lab (*Brady et al., 2013*).

DOI: https://doi.org/10.7554/eLife.45591.002

measures of memory error in degrees (remembered feature value - encoded feature value). Using these fine-grained memory measures, we test how the content and fidelity of information is bound into a single memory representation, and if these memory processes are supported by flexible engagement of the PM and AT cortico-hippocampal networks (*Ritchey et al., 2015a*).

## Episodic features are recollected with varying precision

We first evaluated behavioral performance to quantify memory variability, both in terms of the probability of successful retrieval and precision of each reinstated feature. The proportion of 'correct' responses (memory success) was calculated for each of the object features - item color, spatial context, and emotion association - and the precision of correct retrieval was additionally estimated for color and spatial features. Here memory performance was evaluated by fitting a mixture model (*Bays et al., 2009*; *Zhang and Luck, 2008*) to each participant's response errors (*Figure 2A*; see Materials and methods). Participants remembered the features well above chance on average (*Table 1*), and the proportion of correct responses did not differ between color and scene features (t (27) = 1.09, p=0.29). Participants varied in the precision with which they could remember these visual details, but were more precise when remembering the object's spatial location compared to color (t (27) = 6.48, p<0.001).

## The gist but not precision of episodic features is bound in memory

Based on the hypothesis that interactions between hippocampus and the PM and AT systems support the integration of recollected episodic information, we sought to test if measures of memory success and precision were dependent across features within participants. To this end, we calculated trial-specific measures of memory success (binary correct (1) vs. incorrect (0)) and memory precision

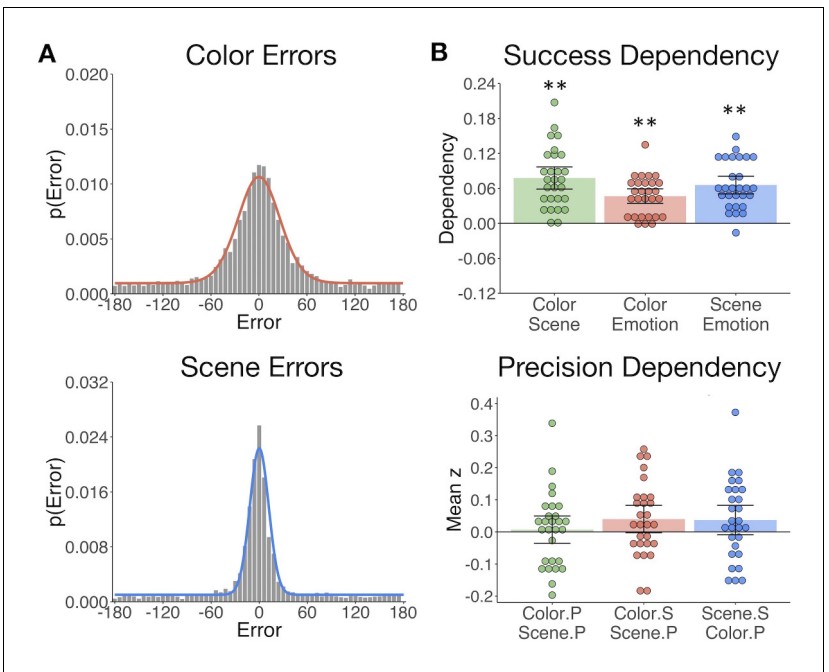

**Figure 2.** The gist but not precision of episodic features is bound in memory. (A) Aggregate color and scene location errors (response - target) with the best-fitting mixture model probability density functions overlaid (*Figure 2—source data 1*). (B) Memory dependency between the features across trials within subjects, in terms of binary 'correct' vs. 'incorrect' retrieval, and the precision of correctly remembered visual information. The top panel shows corrected dependency for successful recall of each feature pair. This measure reflects the observed dependency of each feature pair $[P_{AB} + P_{A'B'}]$ after subtracting the expected dependency from the independent model $[P_A P_B + P_{A'} P_{B'}]$. The bottom panel shows the mean Fisher-transformed Pearson's correlation between the precision (P) of remembered color and scene trials and successful (S; correct vs. incorrect) retrieval of those features (*Figure 2—source data 2*). Bars = Mean + /- 95% CI. **=$p < 0.001$.

DOI: https://doi.org/10.7554/eLife.45591.005

The following source data is available for figure 2:

**Source data 1.** Feature Errors.
DOI: https://doi.org/10.7554/eLife.45591.006
**Source data 2.** Feature Memory Dependency.
DOI: https://doi.org/10.7554/eLife.45591.007

(reversed absolute error of correct trials; see Materials and methods). We expected that successful retrieval of one feature would promote memory for the others (*Horner and Burgess, 2013*;

**Table 1.** Feature memory success and precision.

The proportion of trials for which the emotion, color, and scene features were 'successfully' remembered (note that chance is 0.5 for emotion and 0 for color and scene) and the precision (response concentration $k$) of remembered color and scene features (*Table 1—source data 1*). Means (SE).

| Feature | Memory success | Memory precision |
| --- | --- | --- |
| Emotion | 0.76 (0.02) | — |
| Color | 0.67 (0.04) | 5.40 (0.49) |
| Scene | 0.64 (0.04) | 27.00 (3.30) |

DOI: https://doi.org/10.7554/eLife.45591.003
The following source data is available for Table 1:
**Source data 1.** Feature Memory Success and Precision.
DOI: https://doi.org/10.7554/eLife.45591.004

*Horner and Burgess, 2014*). Successful retrieval must be based on memory for at least the 'gist' of the feature, that is, a general representation that can be high or low in resolution. Therefore, we additionally asked whether successful retrieval further influences the precision with which other visual information is remembered, and if the precision of different features in memory is related. All feature pairs showed significant memory dependency for retrieval success (*Figure 2B* upper panel; ts(27) > 7.30, ps<0.001), so that retrieval of one feature was likely to lead to successful retrieval of the others. However, successful retrieval of color and scene information did not significantly benefit the precision with which the other feature was recalled (ts(27) < 1.85, ps>0.07). Color and scene memory precision were also unrelated (t(27) = 0.32, p=0.75) (*Figure 2B* lower panel). Therefore, integration of episodic information into a coherent memory trace likely involves the binding of gist-like information about distinct features, whereas the specific resolution of each feature in memory appears to be somewhat independent of this binding process.

## Memory retrieval reduces modularity and increases inter-network background connectivity

It remains untested how hippocampal, PM, and AT networks change in their communication during episodic retrieval, and how such changes contribute to episodic memory. Our neuroimaging analyses target this question in a hierarchical manner, testing i) how background network connectivity reorganizes between encoding and retrieval, ii) if dynamic changes in network communication track a measure of multidimensional memory quality, and iii) if dissociable connections support the fidelity of different types of episodic features. We first compared functional connectivity during remember events with connectivity during encoding events. Using the CONN toolbox (*Whitfield-Gabrieli and Nieto-Castanon, 2012*), HRF-weighted correlations between each ROI (*Figure 3A*) times series were computed across encoding and remember task events after first regressing out all trial- and memory-related activity and nuisance variables such as motion (see Materials and methods). Thus, connectivity within each task reflects background covariation in ROI activity independent of trial and behavioral factors driving changes in region-specific activity.

Modularity during each task was calculated from each subject's thresholded (r >= 0.25), weighted connection matrix using the Louvain method of community detection. This algorithm calculates a global modularity value (Q), reflecting the degree to which a set of ROIs are functioning as distinct modules. PM and AT systems appeared to be functioning as relatively distinct networks during encoding (*Figure 3C*), but modularity across our ROIs was significantly reduced during episodic retrieval (t(27) = −3.30, p=0.003), suggesting an increase in inter-network communication and a less segregated network structure (*Figure 3B*). To quantify changes in within-network and between-network communication, mean network density (strength of connections) was calculated for all ROI pairs within the same hypothesized network and for all ROI pairs in different networks. Supporting our a priori network structure, ROIs within the same network had substantially stronger connectivity strength than ROIs between networks (F(1,27) = 132.83, p<0.001). Episodic retrieval was accompanied by an overall increase in connectivity strength relative to encoding (F(1,27) = 14.84, p<0.001), although the change in between-network connectivity was disproportionately greater than change in within-network connectivity (F(1,27) = 11.43, p=0.002). Of note, change in modularity between encoding and retrieval and the disproportionate increase in between-network connectivity strength was robust to different thresholds used to define connections (modularity ts >3.18, ps<0.004; network density interaction Fs > 10.47, ps<0.003; see Materials and methods). Therefore, episodic retrieval is characterized by a notable increase in inter-network connections of hippocampus, PM, and AT regions, and a breakdown in a modular network structure, perhaps facilitating integration of different event features during memory reconstruction.

## Dynamic changes in hippocampal-cortical network connectivity predict memory quality

The background connectivity results suggest that episodic retrieval is associated with a less modular hippocampus, PM, and AT network structure, consistent with prior research (*Westphal et al., 2017*). Yet it is unclear whether these changes in network connectivity reflect a general retrieval state or whether they actually support the recovery of complex episodic information. To address this question, we used generalized psychophysiological interaction (gPPI) analyses to measure how effective

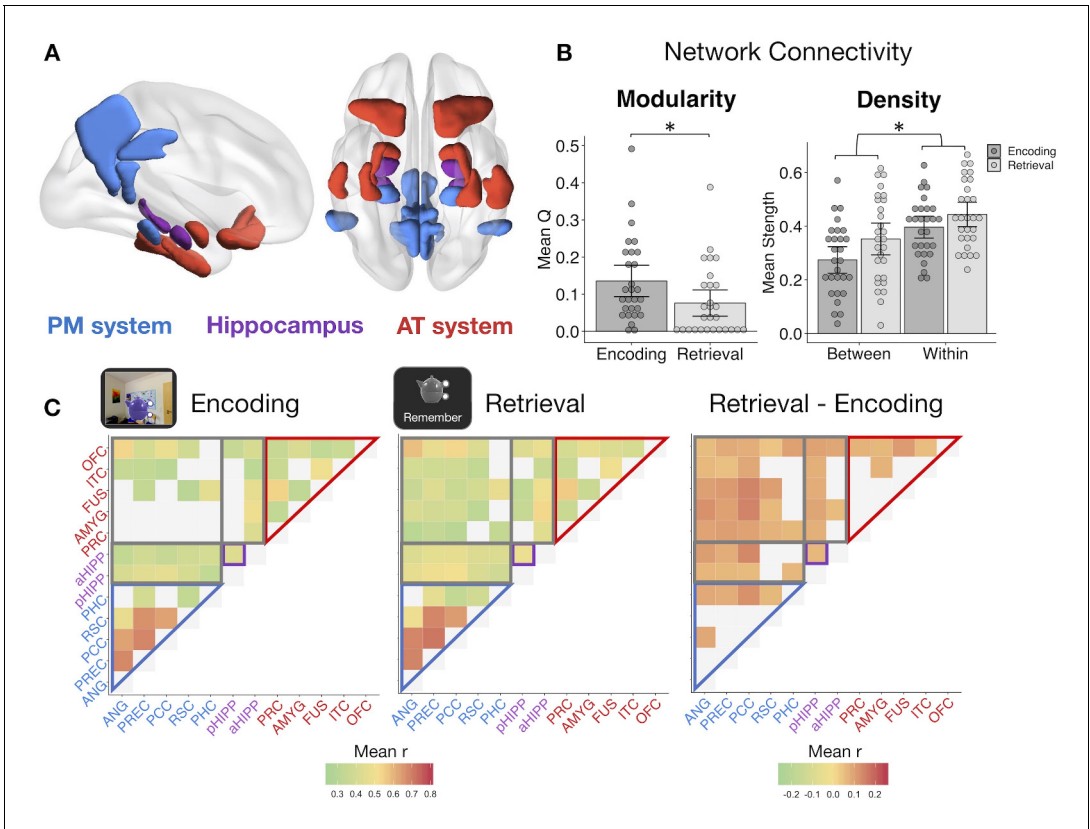

**Figure 3.** Memory retrieval reduces modularity and increases inter-network background connectivity. (**A**) Bilateral anatomical ROIs included in all analyses, obtained from probabilistic atlases in MNI space. PM ROIs: angular gyrus (ANG), precuneus (PREC), posterior cingulate cortex (PCC), retrosplenial cortex (RSC), and parahippocampal cortex (PHC). AT ROIs: perirhinal cortex (PRC), amygdala (AMYG), anterior fusiform gyrus (FUS), anterior inferior temporal cortex (ITC), and lateral orbitofrontal cortex (OFC). Hippocampus was divided into anterior (aHIPP) and posterior (pHIPP). Visualization generated with BrainNet Viewer (*Xia et al., 2013*). (**B**) Mean change in functional connectivity between encoding and retrieval ('remember') events, including overall modularity as well as between- and within-network density (mean strength of connections, defined as r > 0.25) (*Figure 3—source data 1*). Bars = Mean + /- 95% CI, points = individual subject mean estimates. *=*p* < 0.05. (**C**) Mean ROI-to-ROI connectivity during encoding, retrieval, and retrieval - encoding. Connections shown within a task exceed r = 0.25, p<0.05 FDR-corrected, and connections that change between tasks are significantly different from zero, p<0.05 FDR-corrected.

DOI: https://doi.org/10.7554/eLife.45591.008

The following source data is available for figure 3:

**Source data 1.** Network Modularity and Density.
DOI: https://doi.org/10.7554/eLife.45591.009

connectivity of each ROI pair might be modulated by an event-specific, continuous measure of multi-dimensional memory quality. This measure captures fine-grained information bound in memory, accounting for both the amount *and* precision of remembered features (see Materials and methods), thus providing a measure of retrieval sensitive to the quality and diversity of memory content. Note that gPPI measures the influence of a seed on a target region after partialling out task-unrelated connectivity and task-related activity, and thus the results include an asymmetrical effective connectivity matrix.

Averaging across all possible ROI pairs, as predicted, there was an overall increase in connectivity with event-specific increases in memory quality (mean beta = 0.36, SE = 0.16; t(27) = 2.17, p=0.019). Taking the average of within-network and between-network connections for each seed-to-target pair, we next tested how connectivity across our networks changed with increasing quality of remembered details (*Figure 4A*). In line with the results of the background connectivity analyses, it was primarily connections between our networks, particularly with the hippocampus, that increased with memory quality. Specifically, AT-PM connectivity increased with higher memory quality (ts(27) >

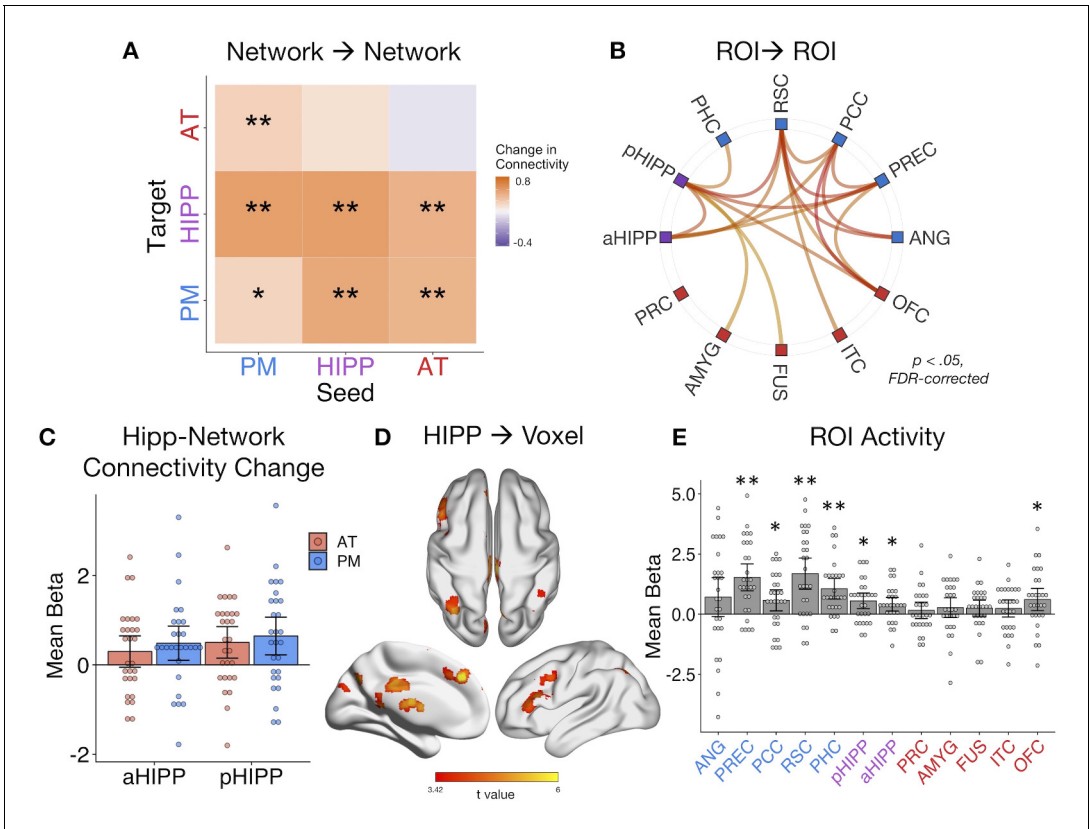

**Figure 4.** Dynamic changes in hippocampal-cortical network connectivity predict multidimensional memory quality. (**A**) Mean change in within- and between-network connection strength with increasing memory quality during remember trials. **=*p* < 0.05, FDR-corrected; *=*p* < 0.05, uncorrected. (**B**) Individual ROI-to-ROI connections whose connectivity strength positively tracks the quality of episodic retrieval. (**C**) Mean change in connectivity between aHipp and pHipp ROIs and regions in the AT and PM systems with increasing memory quality (*Figure 4—source data 1*). (**D**) Hippocampus to voxel connectivity with increasing memory quality. Voxels shown at a peak threshold of p<0.001, and a cluster threshold of p<0.05, FDR-corrected. (**E**) Mean change in bilateral ROI activity with memory quality during retrieval (*Figure 4—source data 2*). **=*p* < 0.001, FDR-corrected; *=*p* < 0.05, FDR-corrected. Bars = Mean +/- 95% CI, points = individual subject estimates.

DOI: https://doi.org/10.7554/eLife.45591.010

The following source data is available for figure 4:

**Source data 1.** Memory-Modulated Hippocampal Connectivity.
DOI: https://doi.org/10.7554/eLife.45591.011
**Source data 2.** Memory-Modulated ROI Activity.
DOI: https://doi.org/10.7554/eLife.45591.012

1.96, ps<0.05, FDR-corrected), and both the AT (t(27) = 3.10, p=0.007, FDR-corrected) and PM (t (27) = 3.29, p=0.007, FDR-corrected) networks increased their connectivity to hippocampus. In tests of the other direction, the hippocampus seed increased its connectivity to the PM system (t(27) = 2.39, p=0.027, FDR-corrected), but not significantly so to the AT system (t(27) = 1.65, p=0.06). This unidirectional hippocampus-AT relationship implies that AT system activity explains more variance in hippocampal activity with greater memory quality, but not vice versa. Turning to within-network connections, anterior and posterior hippocampus increased their connectivity with each other with better memory (t(27) = 3.12, p=0.007, FDR-corrected), but there was only a small change in within-PM connectivity (t(27) = 1.71, p=0.049, uncorrected) and no significant change in within-AT communication (t(27) = 0.73, p=0.235). The individual ROI-to-ROI connections that showed a significant increase in connectivity with memory quality are shown in *Figure 4B*. Of note, when comparing objects that had been associated with an emotionally negative or neutral sound, increases in network connectivity with memory quality appeared to be slightly stronger for negative-associated objects, most predominantly for within-PM connections (t(27) = 2.24, p=0.033 uncorrected), and AT-to-PM

connections (t(27) = 2.74, p=0.011 uncorrected), although these emotion effects did not survive correction for multiple network comparisons.

Exploring these memory-related changes in hippocampal-cortical network connectivity in more detail, we compared anterior and posterior hippocampus: Is there differential connectivity change with the PM and AT systems along the hippocampal long axis? Comparing the mean of bidirectional memory-modulated connectivity between each hippocampal subregion and cortical network revealed no differences between aHipp and pHipp, as well as no differences in connectivity change with the AT and PM systems, and no interaction between these factors (Fs(1,27) < 1.96, ps>0.17) (*Figure 4C*). At the individual region level, there were also no significant differences between pHipp and aHipp in terms of change in connectivity strength with increasing memory quality (|ts| < 2.34, ps>0.26, FDR-corrected). In an exploratory, post-hoc test, we re-ran these analyses using only the most posterior half of pHIPP and the most anterior half of aHIPP to probe if our results might have been a function of how we subdivided the hippocampus. Interestingly, there was still no interaction between hippocampal subregion and network (F(1,27) < 0.01, p=0.94), but across both networks, connectivity changes were significantly greater for pHIPP than aHIPP (F(1,27) = 5.46, p=0.027). Therefore, we found some evidence for differences along the hippocampal long axis, at least in the most extreme segments; compared to aHipp, pHipp exhibited stronger increases in cortical connectivity with higher memory complexity.

Finally, we ran two control analyses to test the role of our ROIs in supporting episodic memory quality. First, to determine whether increases in hippocampal synchrony were specific to our networks of interest or whether evident globally, we analyzed whole-brain connectivity changes with memory. Here, we evaluated the main effect of pHipp and aHipp seeds in terms of the modulatory effect of memory quality on seed-to-voxel connectivity (see *Figure 3D*). The hippocampus increased its communication with voxels in a select group of brain regions, including left dorsolateral prefrontal cortex (−54, 18, 40, k = 784), bilateral parietal cortex (left: −36,−68, 42, k = 440; right: 40,−64, 58, k = 170), precuneus (6,−72,46, k = 437), superior frontal gyrus (multiple clusters, total k = 620), posterior cingulate (4,−28, 30, k = 186), inferior lateral occipital cortex (−50,−76, −20, k = 173), retrosplenial cortex (−2,−42, 2, k = 111), and precentral gyrus (4,−22, 82, k = 90). Second, to verify that our ROIs, particularly hippocampus, showed the expected sensitivity to memory retrieval in our task, we ran a univariate general linear model in which remember events were parametrically modulated by trial-specific values of memory quality. As expected, mean activity of a number of ROIs, particularly within the PM network and hippocampus, positively tracked the quality of episodic retrieval (*Figure 4E*). Surprisingly, unlike other PM regions, the relationship between angular gyrus activity and memory quality was not significant. However, this is likely to be a function of our use of bilateral ROIs to assess network-wide connectivity, where memory effects are more pronounced in left ANG (*Rugg and King, 2018*). The present connectivity analyses control for changes in region-specific activity with memory, thus highlighting the additional importance of functional communication of the PM and AT systems and hippocampus to episodic retrieval.

## Dissociable PMAT connections predict the precision of recalled item and spatial features

The analyses of multidimensional memory quality provide evidence that changes in PM and AT inter-network communication, particularly with hippocampus (cf. *Fornito et al., 2012*; *Geib et al., 2017b*; *King et al., 2015*; *Schedlbauer et al., 2014*), positively track the complexity of information bound within memory. Yet, because this measure is a composite of the quality of all memory features, it remains unknown how PMAT connections support the fidelity of different types of remembered information. This is particularly important to address in light of existing frameworks that emphasize the role of informational content in determining memory organization (*Davachi, 2006*; *Diana et al., 2007*; *Eichenbaum et al., 2012*; *Graham et al., 2010*). In the medial temporal lobes and connected areas (*Ranganath and Ritchey, 2012*; *Ritchey et al., 2015a*), AT regions are sensitive to item-specific associations, and PM regions are sensitive to contextual information, but it is unclear how this organization emerges in terms of network interactions. To this end, we further focused on remember events, specifically trials where a feature was 'successfully' recalled, and tested where changes in connectivity tracked increasing precision of event-specific i) item color and ii) spatial context, given that these measures were found to be independent in memory (see Behavioral Results).

Looking at the average change in connectivity across every seed-to-target pair, we found that there was an overall positive change in connectivity with the precision of both item-color (mean beta = 0.24, SE = 0.11; t(27) = 2.11, p=0.022) and spatial memory (mean beta = 0.25, SE = 0.13; t (27) = 1.94, p=0.032). Are these overall increases in connectivity driven by distinct patterns? Interestingly, within-subject correlations between color and spatial ROIxROI gPPI matrices revealed no evidence for a similar pattern of connectivity changes with memory precision for these features (mean z = 0.02, SE = 0.04; t(27) = 0.50, p=0.312). At the network level (*Figure 5A*), higher color precision in memory was associated with increased connectivity from the AT system to the hippocampus (t(27) = 2.24, p=0.017), between the hippocampus and PM system (ts(27) > 1.82, ps<0.04), as well as increased communication between the AT and PM systems (ts(27) > 2.04, ps<0.026). All other changes in connectivity were not significant (ts(27) < 1.48, ps>.074). Of note, these individual network effects were relatively small and did not survive FDR-correction, although marginal. This may be partially explained by a significant modulatory effect of emotional valence: Objects with a negative association showed more pronounced changes in connectivity with increasing color precision than objects encoded with a neutral sound association (t(27) = 2.34, p=0.027). In contrast to the color precision results, higher spatial precision in memory was accompanied by increased communication strength *within* the PM system (t(27) = 2.88, p=0.018, FDR-corrected) as well as from the AT to the PM system (t(27) = 2.87, p=0.018, FDR-corrected). No other network-level connectivity changes were significant (ts(27) < 1.65, ps>0.06, uncorrected), and these effects were not modulated by the valence of the object's emotion association (t(27) = −0.32, p=0.75). Therefore, item-color

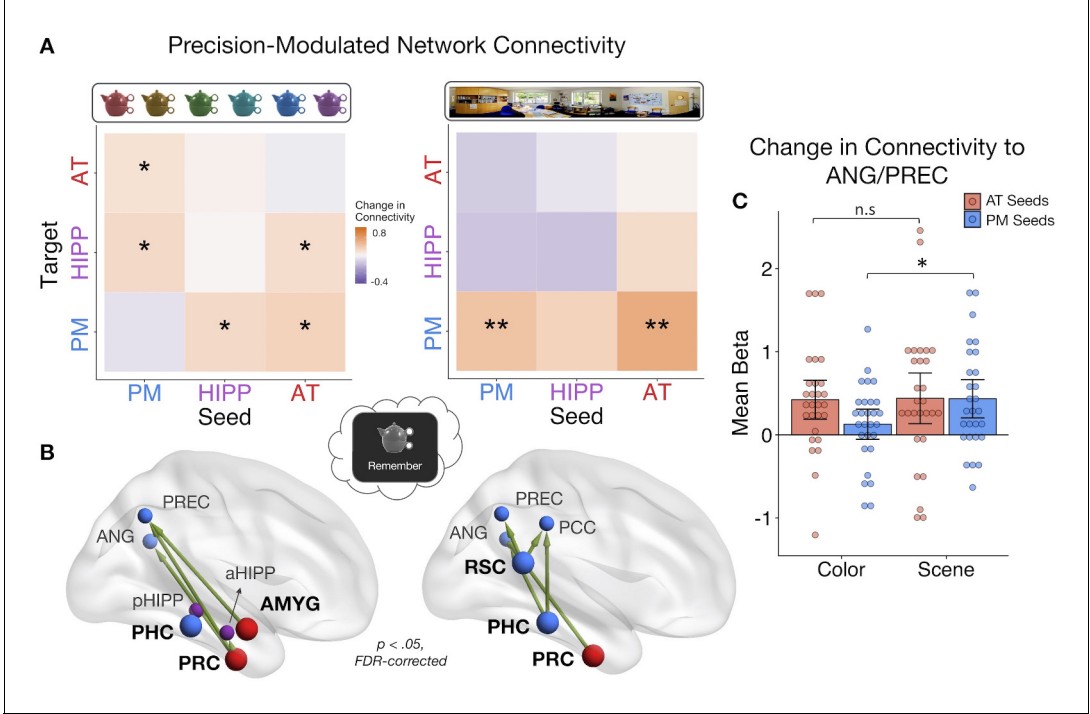

**Figure 5.** PMAT connections predicting the precision of item and spatial features in memory. (**A**) Mean change in within- and between-network connectivity with increasing color memory precision (left) and spatial memory precision (right) during remember trials. \*\*=*p* < 0.05, FDR-corrected; \*=*p* < 0.05, uncorrected. (**B**) Individual seed-to-target connections whose connectivity strength tracks the precision of memory for color (left) and scene (right) information, including PRC and AMYG, sensitive to item and emotion information in the AT system, and PHC and RSC, sensitive to spatial information in the PM system. Depicted connections survive FDR-correction for all possible seed-to-target connections. Seed regions are shown as larger nodes, with bold labels. (**C**) Mean strength of precision-modulated connectivity changes to ANG/PREC for AT seeds (PRC and AMYG) and PM seeds (PHC and RSC), by feature ±95% CI. \*=*p* < 0.05 (*Figure 5—source data 1*). Points = individual subject estimates.

DOI: https://doi.org/10.7554/eLife.45591.013

The following source data is available for figure 5:

**Source data 1.** Feature-Related Connectivity.

DOI: https://doi.org/10.7554/eLife.45591.014

precision and spatial precision in memory were associated with dissociable network connectivity patterns, and these patterns included an increase in inter-network AT connectivity and hippocampal communication for item-color, but an increase in within-PM connectivity and no change in hippocampal communication for spatial information.

Finally, to identify ROI connections that might be contributing to these feature-related network patterns, analyses were further restricted to focus on four seed regions that we hypothesized should show the most representational specificity within our experimental paradigm, including 2 AT regions - PRC and AMYG - and 2 PM regions - PHC and RSC (*Figure 5B*). These regions show the most reliable findings of specificity to (i) item-feature and item-emotion bindings and (ii) spatial representations, respectively (*Epstein, 2008*; *Kensinger et al., 2011*; *Ritchey et al., 2019*; *Robin et al., 2018*; *Staresina et al., 2011*) and are core representational nodes of the PMAT systems (*Ritchey et al., 2015a*). We tested how these seed regions changed their connectivity to all other regions with (i) increasing color memory precision and (ii) increasing spatial memory precision. All statistics were FDR-corrected. For color precision, PRC showed the most widespread changes in connectivity to aHipp, PREC, and ANG (ts(27) > 2.38, ps< 0.045). Additionally, AMYG increased its connectivity with PREC (t(27) = 2.84, p=0.046) and PHC increased its communication with pHipp (t(27) = 2.85, p=0.046). In contrast, the most pronounced increases in connectivity with spatial precision involved PM regions: RSC increased its connection with ANG, PREC, and PCC (ts(27) > 2.91, ps< 0.013), PHC with ANG and PCC (ts(27) > 2.99, ps< 0.016), and PRC increased its communication with ANG (t(27) = 2.91, p=0.040). Because both sets of seeds showed precision-related changes in connectivity with ANG and PREC, two regions previously associated with the vividness and precision of episodic recollection (*Lee et al., 2018*; *Oedekoven et al., 2017*; *Richter et al., 2016*; *Sreekumar et al., 2018*), we ran post-hoc tests to further investigate if PM seed (PHC and RSC) and AT seed (PRC and AMYG) connections to these common targets differed by feature (*Figure 5C*). The change in connectivity of PM seeds to ANG/PREC was significantly greater for spatial than color precision (t(27) = 2.67, p=0.013), but there was no difference between the features in connectivity of AT seeds (t(27) = 0.10, p=0.92). Therefore, although there is notable overlap in the PMAT connections that contribute to the precision of different features in memory, there appears to be a degree of representational specificity in connectivity patterns.

## Discussion

Much research has demonstrated widespread increases in functional connectivity during episodic retrieval (*Fornito et al., 2012*; *Geib et al., 2017a*; *King et al., 2015*; *Schedlbauer et al., 2014*; *St Jacques et al., 2011*; *Westphal et al., 2017*), yet how these changes relate to the phenomenology of recollective experience has remained unknown. The complex process of recollection is associated with several distinct elements, including subjective feelings of vividness, as well as the number of details recalled, types of details recalled, and the precision of that information. Here, we focus on the neural dynamics supporting the content and precision of recollected information. While several prominent accounts have posited that medial temporal regions, including PHC and PRC, provide memories with complementary spatial and item-specific representations, respectively, (*Davachi, 2006*; *Diana et al., 2007*; *Eichenbaum et al., 2012*; *Graham et al., 2010*), a recent model - the PMAT framework - extends this representational sensitivity to large scale cortical networks (*Ranganath and Ritchey, 2012*; *Ritchey et al., 2015a*). Here, a PM system provides the spatial contextual scaffold for event details, including item, emotional, and semantic information provided by an AT system. This content is thought to be integrated as an event via the hippocampus (*Barry and Maguire, 2019*; *Moscovitch et al., 2016*). Although a core prediction of the PMAT framework is that functional *interactions* between cortical systems and the hippocampus are crucial for reinstating multidimensional episodic information, this prediction has not before been tested. First, we found that the PMAT cortical systems functioned in a modular way during memory encoding, with the hippocampus connecting to both systems. In contrast, episodic retrieval was accompanied by a disproportionate increase in inter-network connections. Second, we found that both cortical systems dynamically increased their connectivity to hippocampus with increasing multidimensional quality of episodic memory. Finally, we found that color and spatial memory precision did not clearly map on to changes in AT and PM connectivity, respectively, but rather that feature-related differences emerged in how the networks communicated with each other and with the hippocampus.

Previous research has reliably demonstrated functional segregation of the PMAT systems during rest (*Libby et al., 2012*; *Ritchey et al., 2014*; *Wang et al., 2016*). Here, we observe a similarly clear pattern of modularity with background connectivity during memory encoding, thus extending evidence for this network structure to directed cognition. Interestingly though, this structure was less pronounced during episodic retrieval, which was associated with an increase in between-network connections, supporting the idea that reinstatement of event representations is likely driven by interactions of the hippocampus and cortical regions and between cortical systems. Reduced modularity during memory retrieval has been previously demonstrated at the whole-brain level alongside increased hippocampal connectivity (*Geib et al., 2017b*; *McCormick et al., 2015*; *Westphal et al., 2017*). Enhanced connectivity between large-scale cortical systems, including default mode and frontoparietal control networks, during episodic memory retrieval has also been shown (*Fornito et al., 2012*; *Kragel and Polyn, 2015*; *Robin et al., 2015*; *St Jacques et al., 2011*; *Westphal et al., 2017*). However, none of these prior studies directly compared whole-brain or network-level connectivity during retrieval and encoding. Our finding of greater functional coupling of the PMAT cortical systems and hippocampus during episodic retrieval thus complements this previous work and serves as a necessary foundation for understanding how the retrieval process alters network dynamics.

Extending evidence of PMAT-hippocampal integration during retrieval, we found that connectivity between these networks further tracked the event-specific quality of memory. Previous research has found that increased functional communication, particularly with hippocampus, seems to be important for 'successful' episodic retrieval (*Geib et al., 2017b*; *King et al., 2015*; *Schedlbauer et al., 2014*). Results of our whole-brain connectivity analysis showed that the hippocampus increased its interaction with a select group of regions, most notably posterior medial and left lateral frontal regions, showing a similar pattern to the results of *King et al. (2015)*. In prior studies, memory on each trial has been typically quantified in terms of retrieving or forgetting a single episodic feature or a subjective judgment of recollection, thus neglecting the multidimensional quality of event representations. Here, we used a composite score including information about the number of features present in memory as well as quality of those features. This allowed us to show that inter-network connectivity changes parametrically capture an objective level of memory detail rather than just the process of retrieval. Increased connectivity of PMAT regions to the hippocampus with multidimensional memory quality strongly suggests that hippocampal-cortical connections may specifically act to bind multiple sources of event-specific information together in memory (*Diana et al., 2007*; *Horner et al., 2015*; *Ranganath, 2010*), supporting flexible content retrieval (*Horner and Doeller, 2017*), rather than simply facilitating access to individual associations or providing a general index of recollection. Although we did not find evidence for differences in memory-related network connectivity between our *a priori* anterior and posterior hippocampal subregions, an exploratory analysis revealed differences between the most extreme long axis segments. Posterior hippocampus showed stronger increases in memory-related connectivity than anterior hippocampus, but there were no network-specific differences in connectivity patterns. This result aligns with recent findings of increased representational granularity along the human hippocampal long axis (*Brunec et al., 2018*; *Nadel et al., 2013*; *Poppenk et al., 2013*; *Sheldon et al., 2016*), such that posterior hippocampus might play a greater role in reconstructing detailed, precise episodic information by integrating AT and PM systems.

Our behavioral results additionally revealed new evidence that episodic memories are bound at a level that includes the gist of recovered information, but not necessarily the exact precision with which it is remembered. Successful retrieval of episodic information showed the expected dependent structure of a hippocampal binding process (*Horner and Burgess, 2013*; *Horner and Burgess, 2014*), such that retrieving one feature facilitated memory for the others. This was particularly the case for spatial associations, such that successful retrieval of spatial location was associated with better memory for the other features, supporting the organizational role of space in memory (*Robin, 2018*). Interestingly, the precision of each individual feature was at least partially independent of this binding mechanism, such that retrieving a scene location did not significantly improve the precision of color memory, and vice versa, and the precision of recollected item color and spatial context was also unrelated. These results align with the perspective that the primary role of the hippocampus is to bind event features into a coherent spatio-temporal representation but the quality of individual event features occurs at the level of cortical representations (*Barry and Maguire, 2019*). Therefore, the precision of bound features is theoretically separable from the binding process

itself (cf. *Richter et al., 2016*). Other accounts have emphasized the role of the hippocampus in supporting high-resolution bindings, but these studies have typically focused on the precision of a single association (*Kolarik et al., 2018*; *Nilakantan et al., 2018*; *Yonelinas, 2013*). Thus, the present results question whether there are limits to the precision of hippocampal representations, driven by either the number of multimodal episodic associations or the type or precision of individual associations (*Yonelinas, 2013*).

The current study design additionally allowed us to examine functional connectivity changes associated with the precision of distinct features within memory. We expected that within-PM and PM-hippocampal communication would increase with spatial precision, whereas within-AT and AT-hippocampal communication would increase with item color precision. Our results partially supported these predictions: inter-network connectivity of the AT system, hippocampus, and PM system tracked the precision of item color memory, whereas connectivity to and within the PM system tracked the precision of scene location memory. In line with our hypotheses, connectivity among PHC, RSC and dorsal PM regions scaled with the precision of spatial but not color memory, suggesting that within-PM connectivity might selectively support the resolution of spatial context associations. Much research has documented the complementary roles role of PHC and RSC in spatial processing and navigation (*Epstein, 2008*; *Mitchell et al., 2018*), including sensitivity to distance within virtual environments (*Sulpizio et al., 2014*). Moreover, a recent study showed that RSC is important for forming scene representations and is sensitive to the identity of specific views in 360° panorama scenes (*Robertson et al., 2016*), in line with its role in viewpoint precision demonstrated here. Surprisingly, we found no evidence that hippocampal connectivity supported the precision of PM spatial representations, which is in contrast to evidence implicating the hippocampus, particularly posterior, in spatial precision specifically (*Nadel et al., 2013*; *Nilakantan et al., 2017*; *Nilakantan et al., 2018*; *Stevenson et al., 2018*).

In contrast, color precision was associated with connections between AT regions, particularly PRC, to PM regions and hippocampus. Involvement of the PRC complements previous findings that activity of this region is sensitive to item and item-color bindings in memory (*Diana et al., 2010*; *Staresina and Davachi, 2008*). However, the finding that item-color precision was related to inter-network connectivity, rather than within-AT connectivity, was an unexpected result. There are two possible explanations: First, during episodic reconstruction, the fidelity of item representations may be necessarily integrated within a broader PM contextual framework via the hippocampus. This could explain why hippocampal connectivity supported the precision of color but not necessarily spatial associations in memory. However, color memory precision was not significantly dependent on retrieval of the scene location in our study, providing a tentative argument against this interpretation. Alternatively, the angular gyrus and precuneus may play a content-general role in the retrieval and representation of high-fidelity information, thus explaining increased AT-PM connectivity associated with item-color precision. Previous research has demonstrated consistent involvement of these regions in the representation of subjectively vivid and objectively precise information during memory retrieval using both univariate activation and multivariate methods (*Lee et al., 2018*; *Oedekoven et al., 2017*; *Richter et al., 2016*; *Sreekumar et al., 2018*). Moreover, anterior-posterior neural contributions to memory have been proposed to follow a specificity gradient, from low resolution to high resolution representations respectively, and not strictly based on informational content (*Robin and Moscovitch, 2017*). Our findings lend some support to both perspectives: We find evidence for anterior-posterior content sensitivity in terms of the most influential seed regions, but also common functional projections to angular gyrus and precuneus supporting precise memory retrieval.

The present study revealed network connectivity changes associated with the precision of different features during the same retrieval event, indicating that parallel changes in network dynamics support the complexity and content of episodic memory. Future research should examine the specificity of these cortico-hippocampal connections more closely, for instance, using causal methods that can adjudicate their specific contributions (*Kim et al., 2018*; *Nilakantan et al., 2017*). These methods will be particularly useful given that episodic memories by definition reflect an integrated structure of item and context information. As such, our data show involvement of AT-PM connections, including PRC and PHC seeds, in the precision of both item color and scene location memory, and some prior research has highlighted engagement of PRC and PHC during recall of both object and spatial information (*Burke et al., 2018*; *Ross et al., 2017*). Future research should also account for

the temporal evolution of episodic memory, both in terms of the event itself and the retrieval process. For instance, research has found shifting hippocampal connectivity patterns between memory search and elaboration (*McCormick et al., 2015*; *St Jacques et al., 2011*), and it is an open question how this temporal change would apply to the hippocampal-PMAT connections discussed here. In summary, we provide evidence that PM and AT cortical systems increase their functional communication with each other and hippocampus during episodic retrieval, dynamically increasing with the level of multidimensional memory quality. Moreover, we demonstrate for the first time how these connections support the fidelity of complementary representations, driving the flexible reconstruction of past events.

## Materials and methods

**Key resources table**

| Reagent type (species) or resource | Designation | Source or reference | Identifier | Additional information |
|---|---|---|---|---|
| Software, algorithm | R version 3.5.1, RStudio | R Project for Statistical Computing | https://www.r-project.org/ | |
| Software, algorithm | FMRIPrep v1.0.3 | Poldrack Lab, Stanford University | https://fmriprep.readthedocs.io/en/1.0.3/ | |
| Software, algorithm | MRIQC v0.10.1 | Poldrack Lab, Stanford University | https://mriqc.readthedocs.io/en/0.10.1/ | |
| Software, algorithm | MATLAB 2017a | Mathworks | https://www.mathworks.com/ | |
| Software, algorithm | Psychtoolbox-3 | *Kleiner et al., 2007* | http://psychtoolbox.org/ | |
| Software, algorithm | SPM12 | Wellcome Centre for Neuroimaging, UCL | https://www.fil.ion.ucl.ac.uk/spm/ | |
| Software, algorithm | CONN toolbox v17 | Gabrieli Lab, MIT | https://web.conn-toolbox.org/ | |

### Participants

28 participants took part in the current experiment (16 females, 12 males). All participants were 18–35 years of age (mean = 21.82 years, SD = 3.57) and did not have a history of any psychiatric or neurological disorders. Six additional subjects took part but were excluded from data analyses: two participants did not complete the experiment, one due to anxiety and the other due to excessive movement in the MRI scanner, and four additional participants had chance-level performance on the memory task (based on criteria outlined in Behavioral Analyses). This sample size was selected based on a previous study investigating changes in functional connectivity with memory, also using psychophysiological interaction (PPI) analyses (*King et al., 2015*). Informed consent was obtained from all participants prior to the experiment and participants were reimbursed for their time. Procedures were approved by the Boston College Institutional Review Board.

### Materials

The stimuli used in the current experiment included 144 objects selected from https://bradylab.ucsd.edu/stimuli.html as used in *Brady et al. (2013)*, 12 emotional and neutral sounds selected from the International Affective Digitized Sounds (IADS) database (*Bradley and Lang, 2007*), and six panorama scenes selected from the SUN 360 database (http://3dvision.princeton.edu/projects/2012/SUN360/; *Xiao et al., 2012*).

All of the objects were selected on the basis that they did not have a stereotypical color and were also easily recognizable. 120 unique colors from a continuous color spectrum in CIELAB color space were used to change the appearance of the objects, where each color was separated by three degrees around a 360-degree spectrum. Each object was resized to 240 × 240 pixels when overlaid on a scene and 300 × 300 pixels when presented alone in grayscale. Six of the IADS sounds accompanying the objects were emotionally negative, as defined by valence rating of less than four and an

arousal rating of greater than six on scales of 1 (low) to 9 (high) from the *Bradley and Lang (2007)* norms, and had a mean valence of 2.43 (SD = 0.38) and a mean arousal of 7.63 (SD = 0.35). The six neutral sounds were selected to have a valence between 4.5 and 6.5 and arousal less than 5, with a mean valence of 5.31 (SD = 0.42) and a mean arousal of 4.03 (SD = 0.65). All sounds contained natural, easily recognizable content and were 6 s in duration.

Out of the six panorama scenes used for the experiment, half were indoor locations, including a living room, and office, and a greenhouse, and half were outdoor locations, including a city plaza, a field, and a beach. Each scene was selected through piloting to have no clear areas of symmetry, so that perspectives farther apart, in terms of degrees around panorama, were not obviously more perceptually similar the regions closer together. The original warped panorama images were unwarped to provide naturalistic 100° field-of-view images using the 'pano2photo' function from the SUN 360 database, with each perspective resized to 800 × 600 pixels. Each of the panorama scenes was divided into 120 unique image perspectives, with the center of each perspective shifted by three degrees from the previous.

## Procedures

### Experimental paradigm

The experiment was divided into six study-test blocks, with all phases completed in the MRI scanner. In each study phase, participants completed 24 trials (see *Figure 1A*), each of which began with a 1 s fixation, followed by the presentation of an object-scene-sound event for 6 s. Participants were instructed to remember each object's specific color and location within the panorama scene and were also asked to use the sound to remember the object as a 'bomb' (negative sounds) or as 'safe' (neutral sounds). This instruction encouraged participants to integrate the object and its associated features into a meaningful event. Within a study block, each panorama scene was shown four times and each sound was encoded twice. All objects were trial unique. The object color and scene location values were pseudo-randomly selected with the constraint that objects associated with the same panorama within the same block should be at least 45 degrees apart in their color and location within the scene to minimize interference. The trial order was randomized within each block for every participant. Therefore, across the experiment, participants studied 144 object-scene-sound events, with 72 objects accompanied by negative sound and 72 accompanied by a neutral sound, and 24 objects associated with each of the six panorama scenes. Allocations of the object-color-scene-sound associations were randomly generated for each subject.

In each test phase, participants were tested on their memory for all 24 encoded events. On each trial, a grayscale version of a studied object was shown for 4 s. During this time, participants were asked to recall all of the details associated with that object during the study phase (emotion association, color, and scene location) and to hold that whole image in mind as vividly as possible (see *Figure 1B*). Participants then had an additional 2 s to indicate the object's emotional association. Following a 1 s fixation, participants were then shown the object-scene pairing that they studied, but the object was presented in a random color, in a random location of the associated panorama scene. Participants were asked to reconstruct both the color and scene location of the object as precisely as they could, the order of which was counterbalanced across trials. Participants had up to 6 s to reconstruct each feature, with a 1 s fixation separating these questions. For the 'color' question, participants were instructed to use two button box keys to move counterclockwise or clockwise around the color spectrum to find the color of the object as they studied it originally (target color). For the scene question, participants were asked to move counterclockwise or clockwise around the panorama to find the location in which the object was originally presented (target scene location). The feature value that participants chose for the first question was carried over to the second question. At the end of each test phase, participants were presented with feedback on their performance for 12 s, including the percentage of the time they correctly identified objects as bombs or safe, and the percentage of the time that they were 'close' (defined as ±45 degrees from the target feature value) to the original color or scene location of the objects.

### FMRI data acquisition

MRI scanning was performed using a 3 T Siemens Prisma MRI scanner at the Harvard Center for Brain Science, with a 32-channel head coil. Structural MRI images were obtained using a T1-

weighted (T1w) multiecho MPRAGE protocol (field of view = 256 mm, 1 mm isotropic voxels, 176 sagittal slices with interleaved acquisition, TR = 2530 ms, TE = 1.69/3.55/5.41/7.27 ms, flip angle = 7°, phase encoding: anterior-posterior, parallel imaging = GRAPPA, acceleration factor = 2). Functional images were acquired using a whole brain multiband echo-planar imaging (EPI) sequence (field of view = 208 mm, 2 mm isotropic voxels, 69 slices at T > C−25.0 with interleaved acquisition, TR = 1500 ms, TE = 28 ms, flip angle = 75°, phase encoding: anterior-posterior, parallel imaging = GRAPPA, acceleration factor = 2), for a total of 466 TRs per scan run. Fieldmap scans were acquired to correct the EPI images for signal distortion (TR = 314 ms, TE = 4.45/6.91 ms, flip angle = 55°). Physiological data, including heart rate and respiration, were also collected but were not further analyzed.

## Analyses

### FMRI data preprocessing

MRI data were first converted to Brain Imaging Data Structure (BIDS) format using in-house scripts, verified using the BIDS validator: http://bids-standard.github.io/bids-validator/. MRIQC v0.10.1 (*Esteban et al., 2017*) was used as a preliminary check of MRI data quality. Scan runs were excluded from data analyses if more than 20% of TRs exceeded a framewise displacement of 0.3 mm. Two participants had one scan run excluded using this threshold. A further four participants also successfully completed only 5 out of the six scan runs, three as a result of exiting the scanner early and one due to a technical problem with the sound system during the first run.

All data preprocessing was performed using FMRIPrep v1.0.3 (*Esteban et al., 2019*) with the default processing steps. To summarize: each T1w volume was corrected for intensity non-uniformity and skull-stripped. Brain surfaces were reconstructed using recon-all from FreeSurfer v6.0.0 (https://surfer.nmr.mgh.harvard.edu/). Spatial normalization to the ICBM 152 Nonlinear Asymmetrical template version 2009c was performed through nonlinear registration, using brain-extracted versions of both the T1w volume and template. All analyses reported here use structural and functional data in MNI space. Brain tissue segmentation of cerebrospinal fluid (CSF), white-matter (WM) and gray-matter (GM) was performed on the brain-extracted T1w image. Functional data was slice time corrected, motion corrected, and corrected for field distortion. This was followed by co-registration to the corresponding T1w using boundary-based registration with 9 degrees of freedom. Physiological noise regressors were extracted applying CompCor. A mask to exclude signal with cortical origin was obtained by eroding the brain mask, ensuring it only contained subcortical structures. Six aCompCor components were calculated within the intersection of the subcortical mask and the union of CSF and WM masks calculated in T1w space, after their projection to the native space of each functional run. Framewise displacement was also calculated for each functional run. For further details of the pipeline, including the software packages utilized by FMRIPrep for each preprocessing step, please refer to the online documentation: https://fmriprep.readthedocs.io/en/1.0.3/.

### Regions of interest

Regions of interest (ROIs) included the anterior (head) and posterior (body + tail) hippocampus (aHipp and pHipp, respectively) and regions within the PM system and AT system. PM regions included the parahippocampal cortex (PHC), retrosplenial cortex (RSC), posterior cingulate cortex (PCC), precuneus (PREC), angular gyrus (ANG). AT regions included the perirhinal cortex (PRC), amygdala (AMYG), anterior fusiform gyrus (FUS), anterior inferior temporal cortex (ITC), and lateral orbitofrontal cortex (OFC). The selection of these PM and AT anatomical ROIs was based on previous research demonstrating both resting state and functional separation of these regions into distinct networks (*Libby et al., 2012*; *Ritchey et al., 2014*). All analyses were conducted using the mean voxel value within each bilateral region. ROIs were obtained from probabilistic atlases thresholded at 50%, including a medial temporal lobe atlas (https://neurovault.org/collections/3731/; *Ritchey et al., 2015b*) for hippocampus, PHC, and PRC ROIs, and the Harvard-Oxford cortical and subcortical atlases for all other regions (*Figure 2A*).

### Behavioral analyses

Participants' responses for the item color and scene location questions were analyzed by fitting a mixture model (*Bays et al., 2009*; *Zhang and Luck, 2008*) to errors, both within-participant and

across the aggregate group data. The mixture model includes a uniform distribution to estimate the proportion of responses that reflected guessing, as well as a circular gaussian (von Mises) distribution to estimate the proportion of responses that reflected successful remembering (probability of remembering the target), with some variation in precision:

$$p\left(\hat{\theta}\right) = (1-\gamma)\phi_k\left(\hat{\theta}-\theta\right) + \gamma\frac{1}{2\pi}$$

where, $\hat{\theta}$ represents the reported feature value (in radians), $\theta$ is the target (encoded) feature value, $\gamma$ denotes the proportion of randomly distributed responses, and $\phi_k$ represents the von Mises distribution whose parameters include a mean of zero and concentration $k$, which indicates the precision of responses. Maximum likelihood estimates of $k$ and $\gamma$ were obtained at the group and participant level for each feature type. This model has been used in several previous behavioral and neuroimaging studies to estimate long-term memory performance (*Brady et al., 2013*; *Cooper et al., 2017*; *Murray et al., 2015*; *Nilakantan et al., 2018*; *Richter et al., 2016*; *Xie and Zhang, 2017*). Participants were excluded from all analyses if they had a mean absolute error (response - target) of 75° or more on the color or scene questions, where chance-level performance would result in a mean absolute error of 90°. From pilot work, it was ascertained that such a low level of accuracy resulted in predominantly uniformly distributed data, leading to uninterpretable measures of memory precision and unreliable model estimates of memory performance. Proportion correct was also calculated for the emotion association.

We investigated how the emotion, color, and spatial features were bound in memory within each participant, both in terms of quantity (successful vs. unsuccessful retrieval) and quality (retrieval precision). For the purpose of quantifying trial-specific measures of memory success and memory precision for color and scene features, we used the best fitting mixture model parameters from the aggregate color and scene errors (*Figure 2A*) to generate a remembered vs. forgotten threshold. Specifically, we calculated the probability that each color or scene error fitted the von Mises as opposed to the uniform distribution based on the best fitting probability density function. Errors that had at least a 50% chance of fitting the von Mises component were defined as 'correct' trials, and errors with less than a 50% chance of fitting this component were defined as 'incorrect'. This resulted in a threshold of ±57 degrees for color errors and ±30 degrees for scene errors. First, we computed the trial-to-trial dependency of a binary memory success (correct vs. incorrect) score for each feature pairing (emotion-color, emotion-scene, color-scene), reflecting the proportion of trials where features were remembered or forgotten together:$P_{AB} + P_{A'B'}$. These values were corrected by the level of dependency predicted by an independent model accounting for overall memory accuracy, where better memory would lead to stronger correlations between the feature pairs (*Bisby et al., 2018*; *Horner and Burgess, 2013*; *Horner and Burgess, 2014*):$P_A P_B + P_{A'} P_{B'}$. Second, we calculated the within-participant Pearson's correlation (Fisher z transformed) between the precision of correct color and scene memory and memory success. Trial-specific precision was defined as the reversed absolute error of correct (successfully retrieved) trials, such that higher values reflect higher precision. To test the dependence of color and scene precision, trials were restricted to those where *both* features were successfully recalled. These trial-specific measures of memory success and memory precision were also used to create parametric modulators for fMRI analyses.

## Functional connectivity analyses

All connectivity analyses were conducted using the CONN toolbox (*Whitfield-Gabrieli and Nieto-Castanon, 2012*). In all cases, functional data were first denoised within each scan run, including demeaning, linear detrending, high-pass filtering at 1/128 Hz, and regression of the first principal component from aCompCor - to remove white matter and CSF confounds -, framewise displacement, and six motion parameters. All connectivity estimates were then calculated across the concatenated functional runs, as is standard in CONN. All analyses for hippocampus, PM and AT ROIs used unsmoothed functional data to ensure no voxels were included in mean estimates from outside these anatomical regions. Whole brain analyses used functional data smoothed with a 5 mm FWHM gaussian kernel, masked by gray matter. Connectivity estimates were calculated between the mean time series of each bilateral ROI and then averaged at the network level where applicable.

## Task background connectivity

For analyses of network dynamics during encoding and retrieval, we ran a background connectivity analysis. Here, we first created two task vectors reflecting the occurrence of i) encoding and ii) 'remember' events (the 6 s period containing the grayscale object cue at the beginning of each retrieval trial) within the functional time series. Each event was modeled as a HRF-convolved delta function and all other time points were assigned a value of zero. Five additional parametric covariates were generated for each event type to capture memory effects during encoding and retrieval trials: emotion memory, where trials were coded as incorrect (0), low confidence correct (0.5), or high confidence correct (1), color and scene retrieval success, reflecting binary correct (1) vs. incorrect (0) retrieval, and the precision of 'correct' color and scene memory, coded as the reverse-scored error of successfully remembered trials. Regressors for emotion memory and color and scene retrieval success were mean-centered across all trials within an event (encoding or 'remember'). Regressors for color and scene precision were mean-centered within all successfully remembered trials for that feature. As with the task regressors, all other time points within these memory covariates were then set to zero and the vectors were convolved with the HRF. All task effects and memory covariates were regressed out from the functional data prior to connectivity analyses as part of CONN's denoising step. Therefore, results represent connectivity during encoding and retrieval tasks independent of trial- and memory-related changes in region activity.

To measure connectivity between our ROIs during encoding and retrieval, we calculated the Pearson's correlation between each pair of mean ROI time series weighted by the vectors indicating encoding and remember events. This produced two $12 \times 12$ correlation matrices for each subject - one per task. We computed three measures to compare background connectivity between episodic encoding and retrieval within each subject: (1) Modularity, reflecting the degree to which our regions were operating as distinct networks, computed using the Louvain algorithm from R's NetworkToolbox (*Christensen, 2018*). This method calculates a global modularity value (Q), reflecting the degree to which a set of ROIs are operating as a compartmentalized structure based on their covariation in activity. (2) Within-network density, calculated as the average connectivity strength of all intra-network hippocampus/PM/AT connections, and (3) Between-network density, calculated as the average connectivity strength of all inter-network hippocampus/PM/AT connections. For the purposes of estimating these graph measures, correlation matrices were thresholded at the subject level, with all correlations less than 0.25 set to 0. This threshold was chosen arbitrarily, but note that the pattern of results does not change when using alternative thresholds of 0, 0.1, 0.2, and 0.3 (see Results). For the purpose of evaluating significant connections and changes in individual ROI-to-ROI connections between the tasks, each subject's correlation matrices were Fisher transformed to z scores before averaging at the group-level. These statistics were FDR-corrected for multiple comparisons.

## Memory-modulated connectivity

Generalized psychophysiological interactions (gPPI) analyses (*McLaren et al., 2012*) were used to investigate changes in network connectivity with memory performance from trial-to-trial during remember events. Two models were constructed. The first model tested the modulatory effect of an objective measure of 'multidimensional memory quality' on connectivity. This trial-specific composite memory quality measure incorporated both memory success and memory precision for all three object features. Specifically, memory for each feature (emotional association, item color, scene location) was scaled between 0 (incorrect) and 1 (perfect memory) on each trial. Low confidence, correct memory for the emotion was coded as 0.5 and correct, high confidence emotion memory was coded as 1. Correct memory for color and scene information was further scaled according to the precision of the successfully remembered feature (reversed absolute error), where a value of 1 would reflect perfect feature memory (an error of 0). Finally, these values were summed so that each trial could have a total memory quality score between 0 and 3, with higher values reflecting better memory. Therefore, a maximum value is achieved on any given trial not by simply remembering all features, but by remembering them all with perfect precision. For each participant, this memory quality vector was mean-centered within remember events and convolved with the HRF, with all other time points set to zero. For gPPI analyses, the mean time series of each ROI was predicted by the mean time series of a seed region, a psychological variable containing the HRF-convolved memory quality scores, and the interaction of the seed time series and memory regressor. Taking these interaction

terms produced a 12 × 12 gPPI matrix for each participant reflecting the change in functional connectivity from each seed to target region with higher memory quality (e.g., stronger seed-target connectivity when memory quality is high compared to low). Note that as gPPI measures the task-related change in influence of a seed on a target region after partially out task-unrelated connectivity and task-related activity, the outcome is an asymmetrical effective connectivity matrix.

We then tested whether changing network connectivity might be related to the precision of specific features in memory. In a second model, five parametric modulators captured memory retrieval and precision for the individual episodic features during 'remember' events, as described in Task Background Connectivity: emotion memory (coded in terms of incorrect, low confidence correct, high confidence correct), color and scene retrieval success (coded as binary correct vs. incorrect retrieval), and the precision of 'correct' color and scene memory, coded as the reverse-scored error of remembered trials. In this gPPI analysis, each target ROI time series was predicted by a seed time series, all five memory regressors, and the five seed*memory interaction terms. As before, emotion memory, color and scene memory success regressors were orthogonalized relative to all remember trials, whereas color and scene precision regressors were orthogonalized relative to remember trials where memory for that feature was successfully retrieved. Therefore, each interaction beta reflects the *unique* change in connectivity with each memory measure. Due to the dependency of feature retrieval success and the contrasting independence of feature precision in memory (see Behavioral Results), analyses were focused on changes in connectivity with i) the precision of color information and ii) the precision of spatial information. The output from each gPPI interaction term was a 12 × 12 connectivity matrix containing the beta values for each ROI pair, reflecting the magnitude of connectivity change between two regions with higher memory precision. All gPPI statistics were evaluated using one-tailed tests, due to our hypothesis and prior literature suggesting that better memory is accompanied by increased and not decreased connectivity. All network- and region-level statistics were FDR-corrected for multiple comparisons. Some key scripts and data for behavioral and neuroimaging analyses have been provided here: http://www.thememolab.org/paper-orbitfmri/ (*Cooper and Ritchey, 2019*; copy archived at https://github.com/elifesciences-publications/paper-orbitfmri).

## Acknowledgements

This work was supported by NIH R00MH103401 grant (M.R.). We thank Max Bluestone, Rosalie Samide, and Emily Iannazzi for their assistance with data collection. This research was carried out at the Harvard Center for Brain Science, involving the use of instrumentation supported by the NIH Shared Instrumentation Grant Program; grant number S10OD020039.

## Additional information

### Funding

| Funder | Grant reference number | Author |
| --- | --- | --- |
| National Institutes of Health | R00MH103401 | Maureen Ritchey |

The funders had no role in study design, data collection and interpretation, or the decision to submit the work for publication.

### Author contributions

Rose A Cooper, Conceptualization, Data curation, Software, Formal analysis, Validation, Investigation, Visualization, Methodology, Writing—original draft, Project administration, Writing—review and editing; Maureen Ritchey, Conceptualization, Resources, Supervision, Funding acquisition, Methodology, Writing—original draft, Project administration, Writing—review and editing

### Author ORCIDs

Rose A Cooper  http://orcid.org/0000-0003-1521-8371
Maureen Ritchey  http://orcid.org/0000-0002-5957-3642

## Ethics

Human subjects: Informed consent was obtained from all participants prior to the experiment. Procedures were approved by the Boston College Institutional Review Board (17.026).

## Decision letter and Author response

Decision letter https://doi.org/10.7554/eLife.45591.023
Author response https://doi.org/10.7554/eLife.45591.024

## Additional files

### Supplementary files

• Transparent reporting form
DOI: https://doi.org/10.7554/eLife.45591.015

### Data availability

Data and code have been made available via GitHub: https://github.com/memobc/paper-orbitfmri (copy archived at https://github.com/elifesciences-publications/paper-orbitfmri).

The following previously published datasets were used:

| Author(s) | Year | Dataset title | Dataset URL | Database and Identifier |
|---|---|---|---|---|
| Brady TF, Konkle T, Gill J, Oliva A | 2013 | Visual long-term memory has the same limit on fidelity as visual working memory | https://bradylab.ucsd.edu/stimuli.html | bradylab.ucsd.edu, stimuli |
| Xiao J, Ehinger KA, Oliva A, Torralba A | 2012 | Recognizing Scene Viewpoint using Panoramic Place Representation | http://3dvision.princeton.edu/projects/2012/SUN360/ | 3dvision.princeton.edu, SUN360 |
| Bradley MM, Lang PJ | 2007 | The International Affective Digitized Sounds (2nd Edition; IADS-2) | https://csea.phhp.ufl.edu/media/iadsmessage.html | csea.phhp.ufl.edu, IADS |

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
