## [Decision Letter]

Thank you for submitting your article "Cortico-hippocampal network connections support the multidimensional quality of episodic memory" for consideration by *eLife*. Your article has been reviewed by Laura Colgin as the Senior Editor, a Reviewing Editor, and two reviewers. The reviewers have opted to remain anonymous.

The reviewers have discussed the reviews with one another and the Reviewing Editor has drafted this decision to help you prepare a revised submission.

Summary:

Though features within an event are thought to be integrated to form episodic memories, it is not known how the connectivity between the hippocampus and neocortical regions drive the quality of multidimensional memories. Cooper and Ritchey conducted an experiment whereby participants learned a series of objects consisting of colors, scene locations, and emotions. These features were then tested using continuous measures of memory quality. More specifically, the connectivity between neocortical representations in the posterior-medial/anterior-temporal ("PM/AT") network was examined during encoding and retrieval. Reconstruction of memories during retrieval was associated with increased hippocampal connectivity with other regions that scaled with both the number and quality of features, with dissociable regions representing the precision of individual features. These results have significant implications for our understanding of how memory networks support the retrieval of multidimensional episodic memories in high fidelity.

The authors are to be commended for conducting an extremely rigorous and innovative study, which is expected to have a significant impact on the field. The paper was found to be beautifully written and the analyses, while rigorous and detailed, were found to be presented in an extremely clear and concise manner. However, some major issues were raised that need to be addressed.

Essential revisions:

1) Each event was studied for only 6 seconds. However, during retrieval, participants seem to have much more time to think about the object, as well as to explore each scene during the reconstruction of location (12 seconds in total?). This reconstruction period thus exposes participants to more information and for a longer time than during encoding. Could it be the case that hippocampal networks demonstrate greater connectivity during retrieval because the retrieval period is actually richer in informational content than the encoding period?

2) Clarification is needed for both the calculation of precision and the stochastic dependence of precision. Was it the case that the precision score obtained from the mixture model for each participant was used for further estimates of dependence (i.e. the correlation)? Or was it the case that individual errors that exceeded the "success" criterion calculated from the mixture model were used as the estimate of dependence? The former case would not be problematic, but the latter case may be, depending on how this correlation was conducted.

For example, if many errors per participant were used to derive the measure of dependence, was it the signed or absolute error? A trial with -40 color and +40 location error is identical to a trial with +40 color and +40 location error in terms of quality because the measure wraps around a circle, however there would be no correlation due to the sign. More elaboration of how precision was operationalized is needed here.

3) Related to the above point, did the gPPI measure take into account all the errors from 0-180 along the circle? Or did this measure only account for the "success" trials?

4) More justification/elaboration was needed for the present definition of "gist". From the mixture model analyses conducted in this paper (1 – guess rate), "gist" memories can simultaneously contain low and high resolution features, features that are all high in resolution, and features that are all low in resolution. This is because the stochastic dependence measure of success does not seem to distinguish between whether a feature memory is a high resolution success versus a low resolution success.

5) Related to the above point, reviewers appreciated the inclusion of anterior/posterior HC contributions given increasing interest in functional subdivisions along the long axis of the HC. However, the authors should consider looking at the granularity of representations along the HC long axis in a dimensional way (e.g., Brunec et al., 2018) rather than an anterior/posterior subdivision, as it is likely that the manner in which the HC is subdivided influences the resultant connectivity profiles.

6) The failure to find differences along the hippocampal long axis may also reflect the particular stimuli used in this experiment, rather than the broader/richer spatial and autobiographical categories utilized in other work (e.g., Sheldon et al., 2016), or large-scale virtual reality navigation (e.g., Brunec et al., 2018). This point also relates to the definition of "gist" comment made previously.

7) In Subsection “Dissociable PMAT connections predict the precision of recalled item and spatial features”, the authors present analyses from 4 seed regions of interest. While readers can perhaps intuitively appreciate why the authors focused on these 4 seed regions, it would be helpful to present some rationale for these analyses and the choice of seed regions. For example, why did the authors choose the PHC and RSC over the PCC or ANG? Just one sentence, explaining the motivation for choosing the seed regions (over others) would be helpful.

---

## [Author Response]

Essential revisions:1) Each event was studied for only 6 seconds. However, during retrieval, participants seem to have much more time to think about the object, as well as to explore each scene during the reconstruction of location (12 seconds in total?). This reconstruction period thus exposes participants to more information and for a longer time than during encoding. Could it be the case that hippocampal networks demonstrate greater connectivity during retrieval because the retrieval period is actually richer in informational content than the encoding period?

We thank the reviewers for highlighting that the description of the retrieval phase used for connectivity analyses was unclear. The reviewers correctly note that each event was studied for 6 seconds and that these events were used to estimate encoding-related connectivity. However, for retrieval-related connectivity, we focused on the event at the beginning of each retrieval trial – the ‘remember’ event where participants saw only a grayscale object cue and were asked to covertly reinstate the associated details. Participants saw the grayscale object for 6 seconds before completing the separate color- and scene-specific retrieval events (each an additional 6 seconds). Therefore, the retrieval period of interest is comparable in duration to encoding and actually has less informational content than encoding. We have clarified this in Figure 1 and in subsection “Task Background Connectivity”.

2) Clarification is needed for both the calculation of precision and the stochastic dependence of precision. Was it the case that the precision score obtained from the mixture model for each participant was used for further estimates of dependence (i.e. the correlation)? Or was it the case that individual errors that exceeded the "success" criterion calculated from the mixture model were used as the estimate of dependence? The former case would not be problematic, but the latter case may be, depending on how this correlation was conducted.For example, if many errors per participant were used to derive the measure of dependence, was it the signed or absolute error? A trial with -40 color and +40 location error is identical to a trial with +40 color and +40 location error in terms of quality because the measure wraps around a circle, however there would be no correlation due to the sign. More elaboration of how precision was operationalized is needed here.

We have clarified our description of precision and the dependence measure of precision in the Results section and subsection “Behavioral Analyses”. Dependence was calculated across trials within each participant. To summarize, we calculated a trial-specific measure of precision for successfully remembered (correct) trials. Correct trials were defined based on a threshold obtained from the mixture model fitted to the aggregate group data. Errors that had a least a 50% probability of fitting the von Mises distribution (errors clustered around an error of zero) were defined as ‘correct’ and errors outside of this range defined as ‘incorrect’. Therefore, all color trials that had an absolute error of less than or equal to 57 degrees were defined as correct, whereas all scene trials that had an absolute error of less than or equal to 30 degrees were defined as correct. Precision was defined as the reversed absolute error, only for correct (successfully retrieved) trials. So, an error of zero would be maximally precise whereas an absolute error of 57 or 30 would be the least precise memory for color and scene features, respectively. Only absolute error was used for all analyses because, as the reviewer correctly highlights, there is no meaningful difference in precision for an error of +40 and -40, for example. Therefore, precision dependence was calculated within participant as the correlation between absolute errors of color and scene features, for trials where both were successfully recalled.

3) Related to the above point, did the gPPI measure take into account all the errors from 0-180 along the circle? Or did this measure only account for the "success" trials?

The overall memory quality variable used for the first gPPI analysis incorporated both memory success and precision. This was achieved by allocating trials a value of zero if they were incorrect (an absolute error > 57 or > 30 for color and scene, respectively), or their level of precision (reversed absolute error, scaled between 0 and 1) if correct. This has been clarified in subsection “Memory-Modulated Connectivity”. For the second gPPI analysis where we looked at unique contributions of color and scene precision, trials were restricted to those that were ‘successfully’ recalled for each feature.

4) More justification/elaboration was needed for the present definition of "gist". From the mixture model analyses conducted in this paper (1 – guess rate), "gist" memories can simultaneously contain low and high resolution features, features that are all high in resolution, and features that are all low in resolution. This is because the stochastic dependence measure of success does not seem to distinguish between whether a feature memory is a high resolution success versus a low resolution success.

We thank the reviewers for pointing out that the definition of ‘gist’ used here might not be directly comparable to previous studies. When we refer to correct memory for gist, we do not mean to imply that the memory is therefore necessarily low precision in nature. Rather, we mean that at least the correct gist of a feature has been recalled. In other words, retrieving the gist is a prerequisite for further remembering the precision of a feature, such that ‘gist’ reflects the retrieval of a general representation with or without precision. Within those trials where the gist has been correctly recalled, we can then further explore low precision compared to high precision memories. We do not assume that gist and precision are exclusive of each other nor do we assume that gist is based on semantic (vs. perceptual) information. Therefore, we see stochastic dependence between correct retrieval of feature gist, but no additional relationship between the level of precision of those feature representations. This clarification has been added to the Results section.

5) Related to the above point, reviewers appreciated the inclusion of anterior/posterior HC contributions given increasing interest in functional subdivisions along the long axis of the HC. However, the authors should consider looking at the granularity of representations along the HC long axis in a dimensional way (e.g., Brunec et al., 2018) rather than an anterior/posterior subdivision, as it is likely that the manner in which the HC is subdivided influences the resultant connectivity profiles.

We appreciate that functional changes along the long axis of the hippocampus are an increasing area of research in functional neuroimaging, but we believe that probing the continuous granularity of connectivity changes specifically along the hippocampus extends beyond the network-level questions of our paper. However, as highlighted by the reviewers, it is possible that the lack of differences we observed between anterior and posterior hippocampus are a function of the manner in which we a priori divided this region. To this end, we repeated our gPPI memory quality analysis using the most posterior half of pHIPP and the most anterior half of aHIPP (see Author response image 1). Regardless of whether changes are granular or modular in nature, any linear change from anterior to posterior should be reflected in these more extreme segments. Interestingly, even though we still found no interaction between hippocampus region and network (*p* =.94; with regard to the analysis shown in Figure 4B), the main effect of region is now significant (F(1,27) = 5.46, *p* =.027). These findings suggest that there are no network-specific differences in connectivity change with aHIPP vs. pHIPP, but rather there was an overall stronger change in connectivity of pHIPP than aHIPP with increasing memory quality. This offers some support to the literature cited by the reviewers – posterior Hipp might play a greater role in reconstructing detailed, precise episodic information by integrating AT and PM systems than compared to anterior Hipp. This exploratory analysis has been added to the Results section.

**Author response image 1. respfig1:** The most posterior half of pHIPP and the most anterior half of aHIPP.

6) The failure to find differences along the hippocampal long axis may also reflect the particular stimuli used in this experiment, rather than the broader/richer spatial and autobiographical categories utilized in other work (e.g., Sheldon et al., 2016), or large-scale virtual reality navigation (e.g., Brunec et al., 2018). This point also relates to the definition of "gist" comment made previously.

As presented in the preceding response, our new exploratory analysis provides tentative support for changes in connectivity strength, but not pattern, along the hippocampal long axis. It is possible that task differences might account for differences between our results and those of previous studies. However, our task did involve a high spatial component that subjects experienced as relatively immersive due to the panorama environments. With regard to the cited papers, the most notable differences appear to be the type of analysis, where Brunec et al. (2018) focused on changes within the hippocampus and Sheldon et al. (2016) focused on hippocampal activity and connectivity differences between autobiographical and spatial tasks. In contrast, our anterior-posterior comparison assessed network connectivity changes with increasing episodic memory quality. It remains completely possible that underlying contributions of anterior and posterior hippocampus to episodic memory are different but that they do not show qualitatively different patterns of network connectivity within the same task. This is supported by a previous study that found no evidence for differences in whole-brain resting-state connectivity patterns along the hippocampal long axis (Wang et al., 2019). We agree with the reviewers that understanding the nature of changes along the hippocampus long axis and its interactions with neocortical regions is an extremely interesting area to further explore when considering hippocampal contributions to perception and memory. We believe that this is a question in its own right and best suited for a separate paper. These comments have been added to the Discussion section.

7) In Subsection “Dissociable PMAT connections predict the precision of recalled item and spatial features”, the authors present analyses from 4 seed regions of interest. While readers can perhaps intuitively appreciate why the authors focused on these 4 seed regions, it would be helpful to present some rationale for these analyses and the choice of seed regions. For example, why did the authors choose the PHC and RSC over the PCC or ANG? Just one sentence, explaining the motivation for choosing the seed regions (over others) would be helpful.

We thank the reviewers for highlighting the need to further specify the rationale behind our choice of seed regions. We chose the PRC and AMYG due to their roles in binding object-specific features in perception and memory as well as linking items to their emotional salience. We chose the PHC and RSC because they have been widely found to be crucial for intact episodic memory function and exhibit representational specificity to scenes. In contrast, other core memory regions such as PCC, PREC, and ANG appear to play more domain general, or multimodal, roles in memory retrieval. We have added this explanation to the Results section.